# Instance-Level Panoramic Audio-Visual Saliency Detection and Ranking

## ABSTRACT

Panoramic audio-visual saliency detection is to segment the most attention-attractive regions in 360° panoramic videos with sound. To meticulously delineate the detected salient regions and effectively model human attention shift, we extend this task to more fine-grained instance scenarios: identifying salient object instances and inferring their saliency ranks. In this paper, we propose the first instance-level framework that can simultaneously be applied to segmentation and ranking of multiple salient objects in panoramic videos. Specifically, it consists of a distortion-aware pixel decoder to overcome panoramic distortions, a sequential audio-visual fusion module to integrate audio-visual information, and a spatio-temporal object decoder to separate individual instances and predict their saliency scores. Moreover, owing to the absence of such annotations, we create the ground-truth saliency ranks for the *PAVS10K* benchmark. Extensive experiments demonstrate that our model is capable of achieving state-of-the-art performance on the *PAVS10K* for both saliency detection and ranking tasks. The code and dataset will be released soon.

## CCS CONCEPTS

• **Computing methodologies** → **Video segmentation**.

## KEYWORDS

Audio-visual Fusion, Saliency Detection and Ranking, Panoramic Video, Contrastive Learning, Transformer

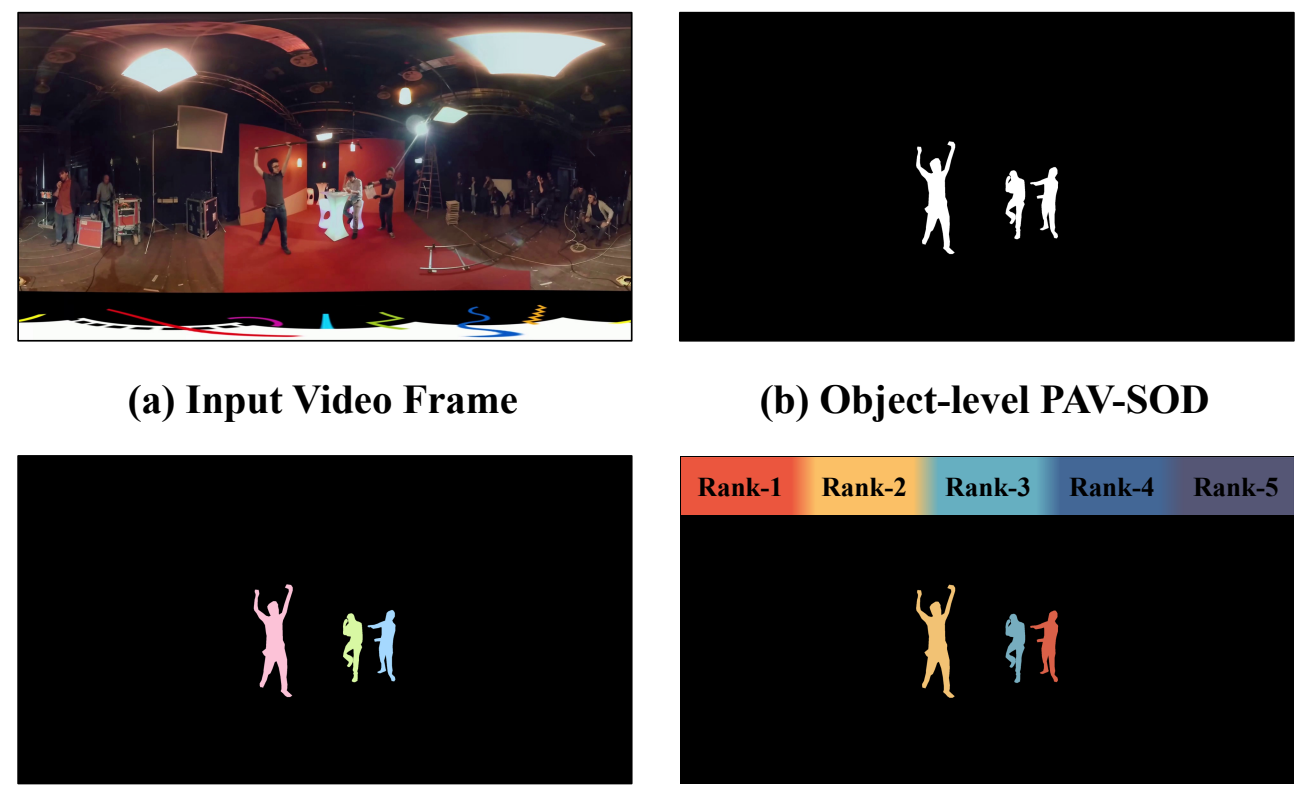

**(a) Input Video Frame** **(b) Object-level PAV-SOD**

**(c) Instance-level PAV-SOD** **(d) Instance-level PAV-SOR**

**Figure 1: Comparison of different panoramic audio-visual saliency tasks. (b) Object-level panoramic audio-visual salient object detection (PAV-SOD) only requires binary segmentation. (c) Instance-level PAV-SOD segments individual salient instances. (d) Instance-level panoramic audio-visual salient object ranking (PAV-SOR) further predicts the relative saliency ranks of different salient instances. Different colors denote different object instances.**

## 1 INTRODUCTION

Recent years have witnessed a burgeoning interest in audio-visual salient object detection (AV-SOD) [2], with the aim of locating video regions that are noticeable and eye-attracting from both visual and audio sources. To study human attention in 360° panoramic real-life environment, [45] establish the first large-scale 360° video dataset and formulates this task as a pixel-wise binary prediction task. Due to being unaware of individual instances of salient objects, we refer to the task as object-level panoramic audio-visual salient object detection (PAV-SOD) in Fig. 1 (b).

Humans, however, are demonstrated to have the ability to identify object instances in the detected salient regions and shift attention from one instance to another when viewing a video of a complex scene. Modeling this ability is crucial for the understanding of how humans interpret videos, and facilitates a wide range of multi-modal applications, e.g., virtual reality, autonomous driving, robot navigation, robot-human interactions. Following this idea, in this paper, we introduce two novel tasks: instance-level panoramic audio-visual salient object detection and ranking, namely instance-level PAV-SOD and PAV-SOR, as depicted in Fig. 1 (c-d). The former segments individual salient instances, and the latter further predicts the relative saliency ranks of these instances. The goal of the proposed tasks is to perform more detailed parsing within detected salient regions and account for inter-observer variability by assigning confidence to different salient instances, making models mimic human perceptual mechanisms promising. For the above instance-level tasks, three key problems need to be considered:

**(i) How to solve panoramic distortions and perceive complicated geometries?** 360° Panoramic video captures the entire surrounding environment and represents each pixel on a 3D sphere. In general, to facilitate storage and transportation, raw panoramic video is transformed into a regular 2D format via equirectangular projection (ERP). However, ERP exhibits severe geometrical distortions, especially in the polar area. Considering the ERP distortion, [36, 37] utilize the combination of equirectangular and cubemap projection as input to the framework, since the distortion can be removed by converting a single equirectangular image into several perspective ones. Nevertheless, this two-branch architecture has

significantly increased the parameters that incurred high computational complexity. In addition, some works use deformable convolution [44] and MLP [11] to mitigate panoramic distortions. **For this Problem**, we present a distortion-aware transformer, which consists of the following steps: 1) sample a small set of neighbor points around each pixel on cubemap and equirectangular projection; 2) aggregate all neighbor features to output the pixel's de-distorted representation; 3) dynamically adjust neighbor positions with learnable position bias. Due to the combination of equirectangular and cubemap projections, our model can effectively avoid panoramic distortions while mitigating discontinuities on cube maps.

**(ii) How to parse audio signals and fuse audio-visual information?** Audio exhibits high overlapping nature, as multiple objects can make sounds simultaneously. Even worse, these audio signals tend to be similar and indistinguishable when the sounding objects are homogeneous. Furthermore, background interference in panoramic videos is stronger than 2D videos with limited field-of-view, leading to audio-visual misalignment. Traditional approaches [45] directly combine entangled audio features with image embeddings by using a bi-linear layer or other fusion operations. However, they fail to effectively unmix audio and correctly locate the sources of sound in visual space. **For this Problem**, we first use the pre-trained SoundNet [1] to extract audio features, and then adopt a sequential multi-modal fusion strategy to integrate visual and audio features. Specifically, an audio-visual spatial activation module embeds entangled audio information into image features and highlights all sounding regions via element-wise summation and convolution operation. To unmix audio signals, we present audio ProtoNet to map dense ambisonics into multiple audio prototypes by using an MLP and a transformer encoder. Then, an audio-visual instance alignment module, equipped with two cross-modal transformers, aligns audio prototypes with visual objects and performs instance-aware audio-visual fusion. Furthermore, we introduce a contrastive learning scheme in the training to ensure each visual object possesses a unique sound semantic corresponding to itself.

**(iii) How to identify individual object instances and predict their saliency ranks?** Instance-level saliency detection is first proposed by [17], and they propose a three-stage method, including salient region detection, object contour detection, and instance generation. This pipeline is cumbersome and its performance is sensitive to post-processing steps. Recently, CATR [18] adopts a DETR-like architecture for audio-visual segmentation (AVS) and treats audio as the prompt to query and segment all sounding object instances. Unlike the AVS task, PAV-SOD aims to predict subjects' fixations and segment the corresponding salient objects, that is, audio cannot be viewed as the necessary and sufficient condition for judging saliency. Moreover, saliency ranks are affected by various factors like object position, motion, sound, etc., making model prediction more difficult. **For this Problem**, a transformer-based object decoder is introduced to establish spatio-temporal relationships among instances and generate final segmentation masks. Note that audio is not the query, but has been embedded as auxiliary information into the visual features. We also show that saliency ranking can be addressed with the same network. After the object decoder, an MLP is added to predict the saliency score for each instance. Additionally, due to the lack of rank annotation, we provide the ground-truth saliency rank based on the attention shift of multiple observers for the *PAVS10K* dataset. In summary, our contributions can be summarized as:

- To the best of our knowledge, this is the first work proposing a unified framework for the instance-level panoramic audio-visual saliency detection and ranking.
- A distortion-aware pixel decoder is designed to mitigate panoramic distortions by aggregating the features of neighbor points sampled from cubemap and equirectangular projection for each pixel.
- A sequential audio-visual fusion module is presented to activate sounding regions and successively perform instance-aware cross-modal fusion. Besides, we introduce contrastive learning to provide regularization for the audio disentanglement and audio-visual alignment.
- Extensive experiments indicate that our model makes great achievements on the *PAVS10K* dataset, outperforming other SOTA methods. For the ranking task, we provide a new evaluation metric and ground-truth saliency rank annotations.

## 2 RELATED WORK

### 2.1 Video Salient Object Detection

Video salient object detection (VSOD), an extension of image saliency detection, aims to segment the most eye-catching objects in a video sequence. Traditional VSOD methods are usually based on handcrafted features, such as motion boundaries [27], center prior [15], long-term point trajectories [25], etc. Recently, some works employ either ConvLSTM [32], 3D convolution [16] or attention mechanisms [8, 12] to better establish the temporal relation in consecutive frames. Furthermore, optical flow is introduced to capture motion saliency clues and enhance spatio-temporal representation. With this strong prior, optical-flow-based models [14, 49] can easily locate the representative salient objects and have achieved impressive performance in VSOD. However, all the abovementioned methods rely on only visual data for saliency detection, neglecting audio hints that also significantly attract human attention under realistic scenarios. Besides, this is contrary to what psychological studies [23, 24] have found out, i.e., human attention is sensitive and susceptible to both visually salient objects and sonic-emitting entities.

### 2.2 Audio-Visual Salient Object Detection

By imitating human perceptual mechanisms in complicated audio-visual scenes, audio-visual salient object detection (AVSOD) is to locate and segment video regions that are salient in both visual and audio sources simultaneously. The main challenge of AVSOD is how to appropriately fuse audio-visual information and learn consistency across modalities. DAVE [34] adopts a straightforward deep-learning-based plain fusion strategy, which takes audio-visual features as input and then concatenates them before saliency prediction. [5] utilize de-convolution operation for the audio-visual alignment, where 1D audio vector is expanded to be the same size as 2D visual counterpart. This strategy views the audio cues as auxiliary information, with corresponding spatial locations being emphasized through embedded semantic consistency. Recent works [35, 45] apply the bi-linear operation to combine multi-modal features. Such a method can maintain spatial structure well, free from dimension

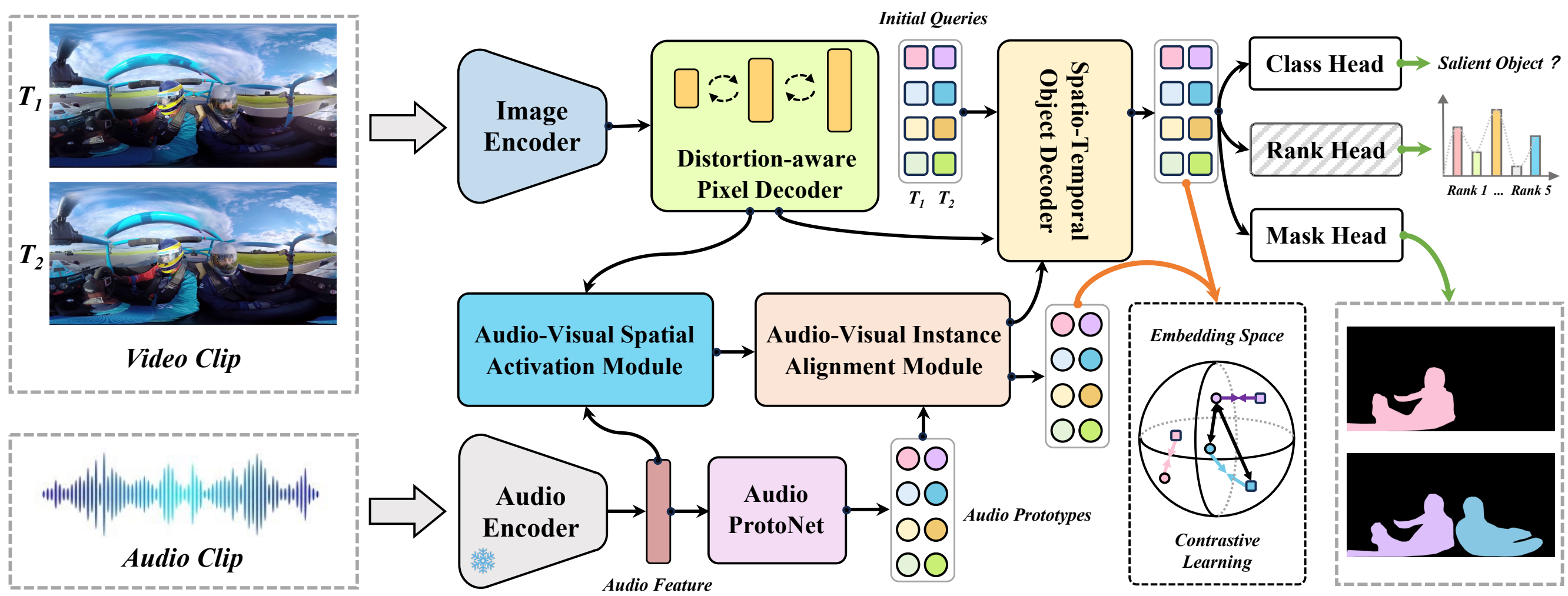

**Figure 2: The overall framework for instance-level panoramic audio-visual saliency detection and ranking. Firstly, the video sequence is divided into T segments, and we use two encoders to extract visual and audio features, respectively. Then, a distortion-aware pixel decoder integrates multi-stage visual features while reducing panoramic distortions via a bi-projection point sampling and a feature aggregation module. Next, we perform audio-visual fusion using the audio-visual spatial activation module and audio-visual instance alignment module sequentially. Finally, a spatio-temporal object decoder takes initial learnable queries and the above cross-modal representations to generate instance-level saliency maps and rank scores.**

mismatching, and enable strong interaction between visual and audio. Moreover, [47] use the self-attention mechanism to build relationships between the visual pixels and the audio signals. This benefits video segmentation with accurate auditory entities while ignoring those spurious sound sources that do not actually emit acoustic signals. [45] propose a new panoramic audio-visual salient object detection (PAVSOD) task, aiming at segmenting the salient objects in 360° panoramic videos. To support the proposed task, they collect the first benchmark *PAVS10K* for PAVSOD and present a baseline model CAV-Net for object-level panoramic saliency prediction by bi-linear fusion and conditional variational auto-encoder. Different from CAV-Net that focuses on detecting pixels belonging to salient regions without considering individual instances, we propose the first pixel-wise instance-level AVSOD method. Instead of just detecting salient regions, our model also distinguishes individual object instances within them. This is essential for real-world applications that require finer distinction.

## 2.3 Audio-Visual Localization and Segmentation

Audio-visual localization [28, 30] aims to locate the regions of sounding objects within the visual frame. Previous methods typically tackle the task through self-supervised or weakly-supervised learning to explore the correlations between audio and visual features, where the goal is to predict the coarse heatmap or bounding box of the sounding object. As a complex extension to the AVL task, audio-visual segmentation [19, 22] is a more challenging task as it requires more fine-grained pixel-level shape description besides localization. Specifically, Zhou *et al.* [48] focus on multi-stage fusion of audio-visual features to facilitate supervised segmentation tasks on their AVSBench dataset, predicting the probability of each pixel in the image belonging to the sounding object. More recently, CATR [18] proposes a decoupled audio-visual transformer that combines audio and video features from temporal and spatial dimensions, capturing their combinatorial dependence. Additionally, they introduce a set of audio-constrained learnable queries to select which object is being referred to segment. In this work, we focus on saliency detection, which imitates human attention and predict subjects' fixations, instead of querying sounding objects. That is, audio cannot be viewed as the necessary and sufficient condition for judging saliency, so this query-based segmentation paradigm is not suitable for saliency detection and ranking tasks.

# 3 METHODOLOGY

Given an input video sequence containing both visual and audio tracks, we first split it into $T$ non-overlapping visual and associated audio segment pairs $\{\mathbf{X^V}_i, \mathbf{X^A}_i\}_{i=1}^{T}$. For each visual snippet, we sample a fixed number of frames. As shown in Fig. 2, we apply an image encoder, ResNet [10], to extract multi-level visual features $\mathbf{f^V} \in \mathbb{R}^{T \times H \times W \times C}$ on video frames. For each audio snippet $\mathbf{X^A}$, we encode it into a feature vector $\mathbf{f^A} \in \mathbb{R}^{D}$ using the first seven 1D convolutions layers of SoundNet [1]. Then, a distortion-aware pixel decoder (Sec. 3.1) is proposed to integrate multi-level visual features while reducing panoramic distortions via a bi-projection point sampling and a feature aggregation module. Next, our model performs audio-visual fusion using the audio-visual spatial activation module and audio-visual instance alignment module sequentially, as detailed in Sec. 3.2. Lastly, a spatio-temporal object decoder (Sec. 3.3) takes initial queries and the above cross-modal features to generate the final salient object instance segmentation masks and the corresponding rank scores.

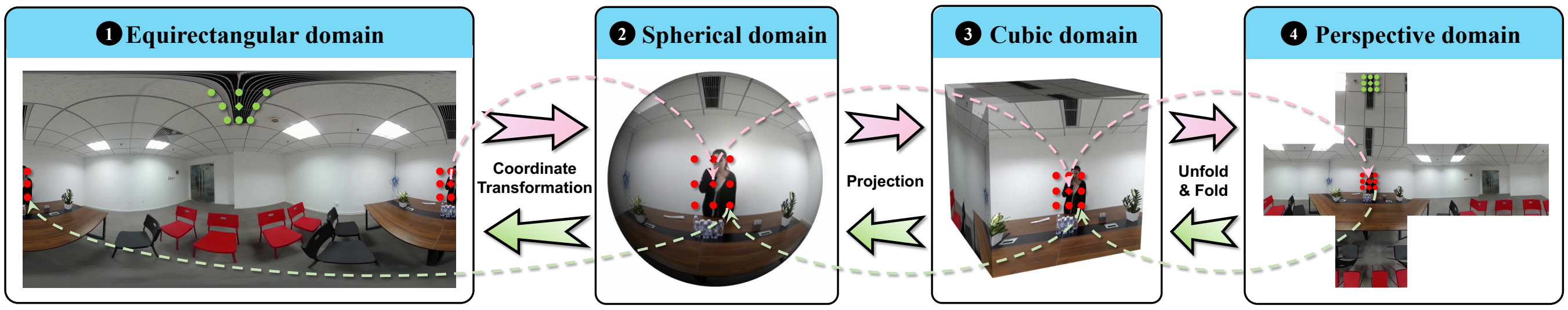

**Figure 3: The cubemap-based sampling method is employed to find neighboring points around each pixel. Firstly, each pixel is projected from the equirectangular domain to the perspective domain. Then, we look up the position of eight nearest neighbors on the cube faces. Lastly, these neighbor points are all projected back to the equirectangular domain.**

### 3.1 Distortion-Aware Pixel Decoder

The equirectangular, commonly used in panoramic videos, is susceptible to geometrical distortions, particularly in the north and south polar regions. To overcome this problem, we present a distortion-aware pixel decoder (DPD) for pixel-wise omnidirectional vision tasks following the three loop steps: 1) sample a small set of initial neighbor points on the cubic and equirectangular domains for each pixel; 2) aggregate all neighborhood features to form the pixel's output feature across multi-scale feature maps; 3) dynamically adjust the position of sampling points with learnable position bias. By aggregating relative neighbor points, our model mitigates panoramic distortions while learning intricate patterns and local region representation, thereby improving performance in panoramic dense prediction tasks.

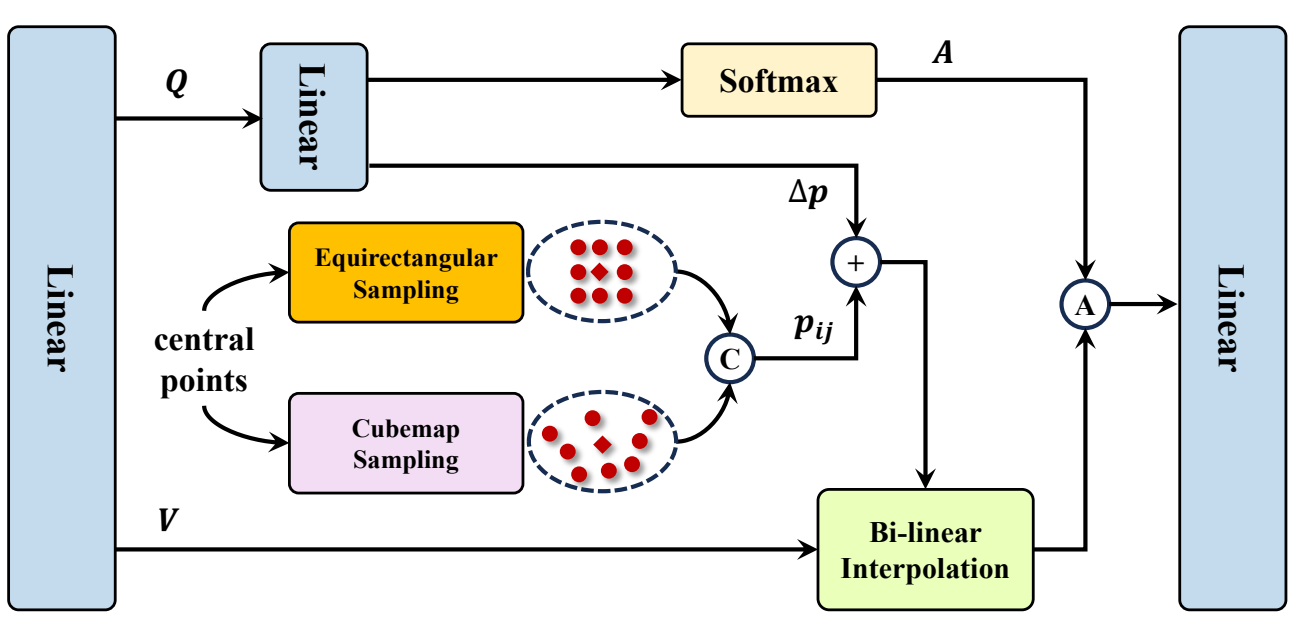

**Figure 4: The distortion-aware aggregation module. This module is the basic building block of the proposed distortion-aware pixel decoder. It takes central points and image feature maps as input, and then learn the neighbor positions of each central point. Finally, we aggregate all neighbors' features and output the final feature of each central point, thereby reducing the negative effect of distortions and perceive more complicated geometric structures.**

3.1.1 *Bi-Projection Neighbor Point Sampling.* Given the input image feature map $\mathbf{f}^{\mathbf{V}}$, let $p_i$ be a pixel in the feature map, namely central point, and $p_{ij}$ refers to its neighbor point. We adopt a bi-projection neighbor point sampling scheme to find neighbor points around each central point by leveraging two projections: equirectangular (ER) and cubemap projections. Specifically, cubemap projection avoids distortion but incurs discontinuity at the cube boundary, whereas ER projection incorporates a complete field-of-view but introduces distortion. To mitigate distortions and ensure continuity simultaneously, we sample neighbors on both the cubic and ER domains. For ER sampling, we directly select the eight nearest neighbor pixels of each pixel on the ER projection as neighbor points. The process is given as follows:

$$p_{ij} = p_i(x \pm a, y \pm b), \{a, b = 0, 1; a, b = 0, 2\} \tag{1}$$

where $a$ and $b$ are the sampling step size. For cubemap sampling, it contains the following steps: ***i) equirectangular-to-cube transformation.*** Let the side length of a cube map be $w$. As the field-of-view (FoV) of each face is 90°, each face can be treated as a perspective camera with a focal length of $\frac{w}{2}$, and they all share the same center point in the world coordinate system. Due to the fixed viewing direction in cubemap projection, a rotation matrix $R_h$ can represent the extrinsic matrix of each camera. For a pixel $p_i$ on equirectangular map, we can transform it into the coordinate on the certain cube face $h$ by the following mapping:

$$q_i^x = \sin(\theta) \cdot \cos(\phi); q_i^y = \sin(\phi); q_i^z = \cos(\theta) \cdot \cos(\phi)$$
$$K = \begin{bmatrix} w/2 & 0 & w/2 \\ 0 & w/2 & w/2 \\ 0 & 0 & 1 \end{bmatrix}; \hat{p}_i = K \cdot R_h^T \cdot q_i \tag{2}$$

where $\theta$ and $\phi$ represent the longitude and latitude of point $p_i$ on the sphere, respectively. The range of $\theta$ spans from $-\pi$ to $+\pi$, while the range of $\phi$ spans from $-0.5\pi$ to $+0.5\pi$. The x, y, and z components of vector $q_i$ are represented as $q_i^x$, $q_i^y$, and $q_i^z$, respectively. ***ii) uniform sampling on the cube map.*** Similar to ER sampling, we select the eight nearest neighbor pixels of each central pixel in the perspective domain, as shown in Fig. 3. ***iii) cube-to-equirectangular transformation.*** All these neighbors are projected back to the ER domain. Given a neighboring point $p_{ij}$ on a specific face $h$, we can perform a coordinate transformation to map it onto the ER projection using the following mapping:

$$q_{ij} = R_i \cdot K^{-1} \cdot \hat{p_{ij}}$$
$$\theta = \arctan\left(q_{ij}^x / q_{ij}^z\right); \phi = \arcsin\left(q_{ij}^y / |q_{ij}|\right) \tag{3}$$

Take $15 \times 30$ feature map as an example, Fig. 8 visualizes all the central points and their neighbor points that are projected from cube map to equirectangular map.

*3.1.2 Multi-scale Distortion-Aware Aggregation.* As illustrated in Fig. 4, image feature maps $\mathbf{F}^{\mathbf{V}}$ are flattened and reshaped into a sequence, and then a linear projection is applied to produce query $\mathbf{Q}$ and value $\mathbf{V}$, inspired by [50]. Next, we sample two groups of neighbor points for each central point via the bi-projection neighbor point sampling strategy, and combine them as initial sampling points $\mathbf{p_{ij}}$. The $\mathbf{Q}$ is passed into two linear layers for calculating neighbor scores $\mathbf{A}$ after a softmax operator and the neighbor offsets $\Delta\mathbf{p}$. The initial neighbor points $\mathbf{p_{ij}}$ and $\Delta\mathbf{p}$ are added together to obtain the final neighbor points. By dynamically adjusting the spatial distribution of sampling points, our model can capture various geometric characteristics and perceive more complicated patterns. Moreover, we adopt bi-linear interpolation on $\mathbf{V}$ to generate neighbor features and then aggregate them using neighbor scores $\mathbf{A}$. Lastly, a linear projection is used to output distortion-aware visual features. Note that we perform distortion-aware aggregation on multi-stage features with resolution 1/8, 1/16, and 1/32 at once.

## 3.2 Sequential Audio-Visual Fusion

Our objective is to fuse visual and audio features, thereby retrieving salient objects from both modalities. To achieve this, we propose a sequential multi-modal fusion scheme with two components: 1) an audio-visual spatial activation module to embed entangled audio information into visual features and activate all sounding regions; 2) an audio ProtoNet to unmix dense ambisonics into a set of prototypes, and then an audio-visual instance alignment module to assign them into individual visual instances and capture instance-aware audio-visual dependence by introducing contrastive learning and cross-modal transformer, respectively.

*3.2.1 Audio-Visual Spatial Activation Module.* As shown in Fig. 2, audio-visual spatial activation module (AV-SAM) takes audio features $\mathbf{f}^{\mathbf{A}}$ and distortion-aware image features $\mathbf{f}^{\mathbf{D}}$ from DPD as inputs. Specifically, $\mathbf{f}^{\mathbf{A}}$ is first spatially duplicated $THW$ times and projected to the same size as $\mathbf{f}^{\mathbf{D}}$. Then we add the above features of two modalities and apply $3 \times 3$ convolution to locate sounding regions, which can be formulated as:

$$\mathbf{f}_{\mathbf{i}}^{\mathbf{SA}} = Conv(\mathbf{f}_{\mathbf{i}}^{\mathbf{D}} + \mathrm{copy}_{1\times1\rightarrow H\times W}(\mathbf{f}_{\mathbf{i}}^{\mathbf{A}})), \forall i \in \{1 \dots T\} \qquad (4)$$

In this way, 1D audio features are treated as auxiliary information and correlated to each spatial location of the visual counterpart. This simple multi-modal fusion strategy can well retain 2D spatial structure and activate all sounding objects.

*3.2.2 Audio-Visual Instance Alignment Module.* While AV-SAM can exploit audio information and build inter-modal relations, the interaction between visual and audio is still weak and ambiguous. To better learn the matching relationship between each visual object and sounds in complicated scenarios, we propose an audio ProtoNet and an audio-visual instance alignment module (AV-IAM), as depicted in Fig. 2. In audio ProtoNet, we first use an MLP to transform audio feature $\mathbf{f}^{\mathbf{A}}$ into multiple implicit sub-variables (referred to as audio prototypes) $\mathbf{p}^{\mathbf{A}} \in \mathbb{R}^{(T\times N)\times C}$, where $N$ is the number of the object query. Next, an original transformer encoder [6] is adopted to perform self-attention on $TN$ audio prototypes as:

$$\hat{\mathbf{p}}^{\mathbf{A}} = SelfAttn(\mathrm{reshape}_{T\times NC\rightarrow TN\times C}(MLP(\mathbf{f}^{\mathbf{A}}))) \qquad (5)$$

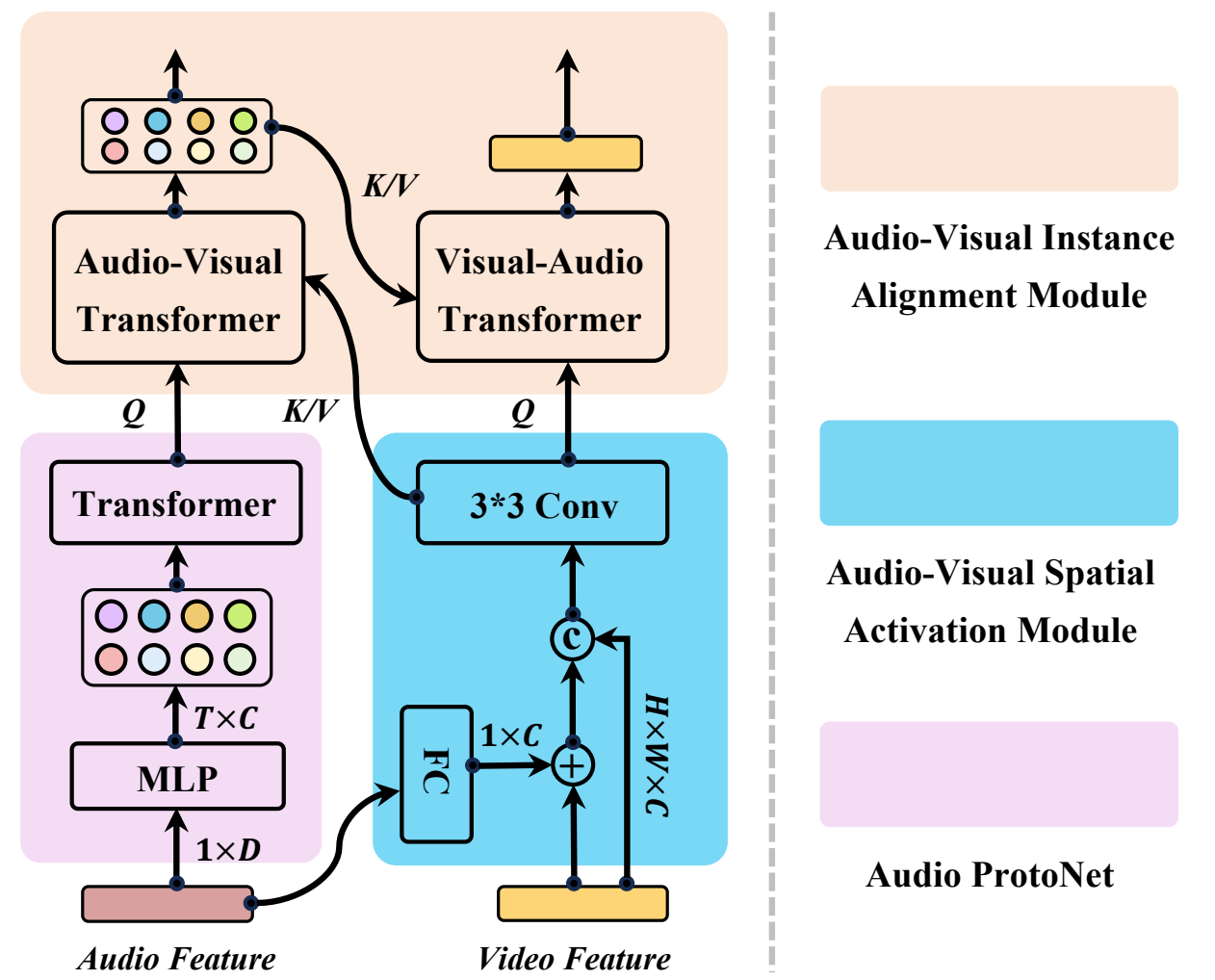

**Figure 5: The sequential audio-visual fusion module. It consists of two main components: 1) an audio-visual spatial activation module (blue) to embed entangled audio information into visual features and activate all sounding regions; 2) an audio ProtoNet (red) to unmix dense ambisonics into a set of prototypes, and then an audio-visual instance alignment module (orange) to assign them into individual visual instances and capture instance-aware audio-visual dependence.**

In AV-IAM, audio prototypes $\hat{\mathbf{p}}^{\mathbf{A}}$ and AV-SAM features $\mathbf{f}^{\mathbf{SA}}$ are passed as input to an audio-visual transformer, where the query is $\mathbf{p}^{\mathbf{A}}$ and the key/value is $\mathbf{f}^{\mathbf{SA}}$. Consequentially, the audio-visual transformer produces audio-conditioned image attention and a-to-v representations in the audio stream, so as to associate each audio prototype with different sounding objects:

$$\mathbf{f}^{\mathbf{AV}} = CrossAttn(\hat{\mathbf{p}}^{\mathbf{A}}, \mathbf{f}^{\mathbf{SA}}, \mathbf{f}^{\mathbf{SA}}) \qquad (6)$$

In order to ensure the disentanglement of audio representations and align the corresponding audio prototypes and pixel features, we introduce an audio-visual contrastive learning objective after the audio-visual transformer. To be specific, we measure the similarity between a-to-v representations $\mathbf{f}^{\mathbf{AV}}$ and query features $\mathbf{f}^{\mathbf{Q}}$ from spatio-temporal object decoder by calculating cosine distance between them in embedding space, where the cosine distance function is defined as $s(u, v) = u^T v/\|u\|\|v\|$. We would like to enforce the consistency between modalities by pulling close the query features whose category is the salient object with its corresponding audio prototypes. To increase the heterogeneity between audio prototypes, we push away the above audio representations. Therefore, by implementing the NT-Xent loss [3], our audio-visual contrastive loss between $\mathbf{f}^{\mathbf{AV}}$ and $\mathbf{f}^{\mathbf{Q}}$ can be formulated as follows:

$$\mathcal{L}_{cl} = -\log \frac{\exp\left(s\left(\mathbf{f}_{\mathbf{n}}^{\mathbf{Q}}, \mathbf{f}_{\mathbf{n}}^{\mathbf{AV}}\right)/\tau\right)}{\sum_{\substack{k=1\\k\neq n}}^{N} \exp\left(s\left(\mathbf{f}_{\mathbf{n}}^{\mathbf{Q}}, \mathbf{f}_{\mathbf{k}}^{\mathbf{Q}}\right)/\tau\right) + \sum_{k=1}^{N} \exp\left(s\left(\mathbf{f}_{\mathbf{n}}^{\mathbf{Q}}, \mathbf{f}_{\mathbf{k}}^{\mathbf{AV}}\right)/\tau\right)} \qquad (7)$$

where $\forall n, \forall k \in \{1 \dots N\}$. $\tau$ is a temperature co-efficient and $N$ is the number of object query. For an audio prototype $\mathbf{f}_{\mathbf{n}}^{\mathbf{Q}}$, the corresponding $(\mathbf{f}_{\mathbf{n}}^{\mathbf{Q}}, \mathbf{f}_{\mathbf{n}}^{\mathbf{AV}})$ is regarded as a positive pair, and the rest $N-1$ audio prototypes constitute negative examples. We intend to

maximize the similarity of positive pairs while minimizing the similarity of negative pairs in embedding space. This process enables our model to disengage dense audio signals into unique sounding embeddings. After that, a visual-audio transformer treats $\mathbf{f}^{\mathrm{SA}}$ as query and a-to-v representation as key/value, and computes cross-attention on them to retrieve the audio prototype for each pixel. This instance-aware alignment method allows our model to embed disentangled audio representations into individual visual objects and further guide the proposed instance-level saliency detection and ranking tasks. The process can be defined as:

$$\mathbf{f}^{\mathrm{VA}} = CrossAttn(\mathbf{f}^{\mathrm{SA}}, \mathbf{f}^{\mathrm{AV}}, \mathbf{f}^{\mathrm{AV}}) \tag{8}$$

### 3.3 Spatio-Temporal Object Decoder

Spatio-temporal object decoder (STOD) aims to predict salient object instance masks for each frame. Motivated by [38], we also initialize a fixed number of learnable positional embeddings, termed object queries. Assuming the model decodes $N$ instances per frame, the total number of object queries for $T$ frames becomes $T \times N$, which corresponds to the size of the audio embeddings. All queries are fed into STOD along with the AV-IAM features. Similar to [4], STOD consists of two successive attention modules: 1) masked self-attention is performed among all queries from $T$ frames to integrate temporal information; 2) cross-attention is computed between object queries and v-to-a features from the visual-audio transformer to generate query features $\mathbf{f}^{\mathrm{Q}}$ over space. Ultimately, two linear classifiers, i.e., class head and rank head, determine whether each object query is a salient object and output the corresponding rank order. For mask prediction, a dot product followed by a sigmoid activation is applied between the query features and DPD's features to produce the final masks.

In the training phase, we assign a unique ground truth for each instance prediction. The matching process can be done through one-to-one bipartite matching strategy. To find the best assignment of a prediction to ground truth, we uniformly sample a set of points on the predicted mask and then construct a cost matrix. Given a matching, each prediction is supervised with a saliency classification loss and a mask loss. The former is binary cross-entropy loss and the latter is composed of a focal loss and a dice loss. The final loss function can be formulated as:

$$\mathcal{L} = \lambda_{ce}\mathcal{L}_{ce} + \lambda_{focal}\mathcal{L}_{focal} + \lambda_{dice}\mathcal{L}_{dice} + \lambda_{cl}\mathcal{L}_{cl} \tag{9}$$

## 4 EXPERIMENTS

### 4.1 Dataset and Evaluation Metrics

We evaluate the performance of our method on the *PAVS10K* dataset [45], comprising 40 training videos totaling 5,796 frames and 27 testing videos with 4,669 frames. These videos are randomly selected from 67 4K-resolution equirectangular videos (each of 30 seconds), following an approximate 6:4 ratio. Instance-level pixel-wise masks have been provided in the original *PAVS10K* dataset. For the PAV-SOR, we use the rank labeling method proposed in [31] to generate our ground-truth saliency rank annotations. Quantitative results on 6 widely used evaluation metrics are reported: F-measure, E-measure, S-measure, MAE, SOR, and #images used.

### 4.2 Implementation Details

We use PyTorch and follow CAV-Net [45] setting for instance-level PAV-SOD and PAV-SOR. The shorter side of all frames is resized to 480 during training and evaluation, without relying on multi-scale or any other data augmentation strategies. Our model is optimized by AdamW algorithm with weight decay 0. By setting the default video clip length as 2 and batchsize as 2, the training of the entire framework starts with an initial learning rate of $1e-4$ on a single NVIDIA Quadro RTX 6000 GPU.

### 4.3 Panoramic Audio-Visual Saliency Detection

To verify the effectiveness of our model on PAV-SOD, we compare our model with the state-of-the-art methods, including Image SOD (I.): CPD-R [41], SCRN [42], F3Net [39], MINet [26], LDF [40], CSFR2 [7], GateNet [46], Video SOD (V.): COSNet [20], RCRNet [43], PCSA [9], 3DC [21], RTNet [29], Panoramic Image SOD (PI.): FANet [13], and Panoramic Audio-Visual SOD (PAV.): CAVNet.

**Table 1: Quantitative comparison with different SOD methods on the F-measure ($F_\beta$), E-measure($E_\epsilon$), S-measure($S_\alpha$) and MAE ($\mathcal{M}$) metrics. These object-level (Obj.) approaches and our instance-level (Ins.) model are equipped with the ResNet-50 backbone. Ours and Ours† are the PAV-SOD and PAV-SOR models, respectively.**

| Level | Type | Model | $F_\beta \uparrow$ | $S_\alpha \uparrow$ | $E_\epsilon \uparrow$ | $\mathcal{M} \downarrow$ |
|---|---|---|---|---|---|---|
| Obj. | I. | CPD-R [41] | 24.3 | 60.9 | 64.8 | .026 |
| | | SCRN [42] | 28.6 | 65.5 | 64.1 | .034 |
| | | F3Net [39] | 31.0 | 64.2 | 69.1 | .029 |
| | | MINet [26] | 28.6 | 62.4 | 65.2 | .044 |
| | | LDF [40] | 32.2 | 64.5 | 70.1 | .035 |
| | | CSFR2 [7] | 29.0 | 64.6 | 68.4 | .026 |
| | | GateNet [46] | 27.3 | 65.3 | 63.6 | .033 |
| | V. | COSNet [20] | 18.1 | 58.2 | 55.9 | **.023** |
| | | RCRNet [43] | 31.0 | 65.4 | 68.8 | .029 |
| | | PCSA [9] | 18.4 | 60.0 | 57.0 | .027 |
| | | 3DCSeg [21] | 30.0 | 64.0 | 60.8 | .055 |
| | | RTNet [29] | 24.7 | 59.1 | 68.3 | .029 |
| | PI. | FANet [13] | 24.1 | 59.6 | 65.4 | **.025** |
| | PAV. | CAVNet [45] | 32.5 | 63.3 | 69.8 | .027 |
| Ins. | PAV. | Ours | **43.2** | **69.9** | **74.0** | .033 |
| | | Ours† | **43.6** | **70.2** | **74.4** | .028 |

The quantitative results are shown in Table 1. For main metrics, our model has obvious improvement among various approaches. Specifically, it reaches 43.2% on $F_\beta$, 69.9% on $S_\alpha$, and 74.0% on $E_\epsilon$, surpassing the second-best ones with 10.7%, 6.6%, and 4.2%, respectively. When introducing ranking supervision, our model exhibits superior results, suggesting that the newly incorporated rank prediction branch can augment the performance of PAV-SOD task. Additionally, our model is the first one that addresses instance-level saliency detection while the existing ones are object-level.

The qualitative results are shown in Fig 6. We compare our model with CAVNet on *PAVS10K* dataset and the result indicates that our model performs better in various challenging cases based on the following fundamental observations: 1) locate the correct salient

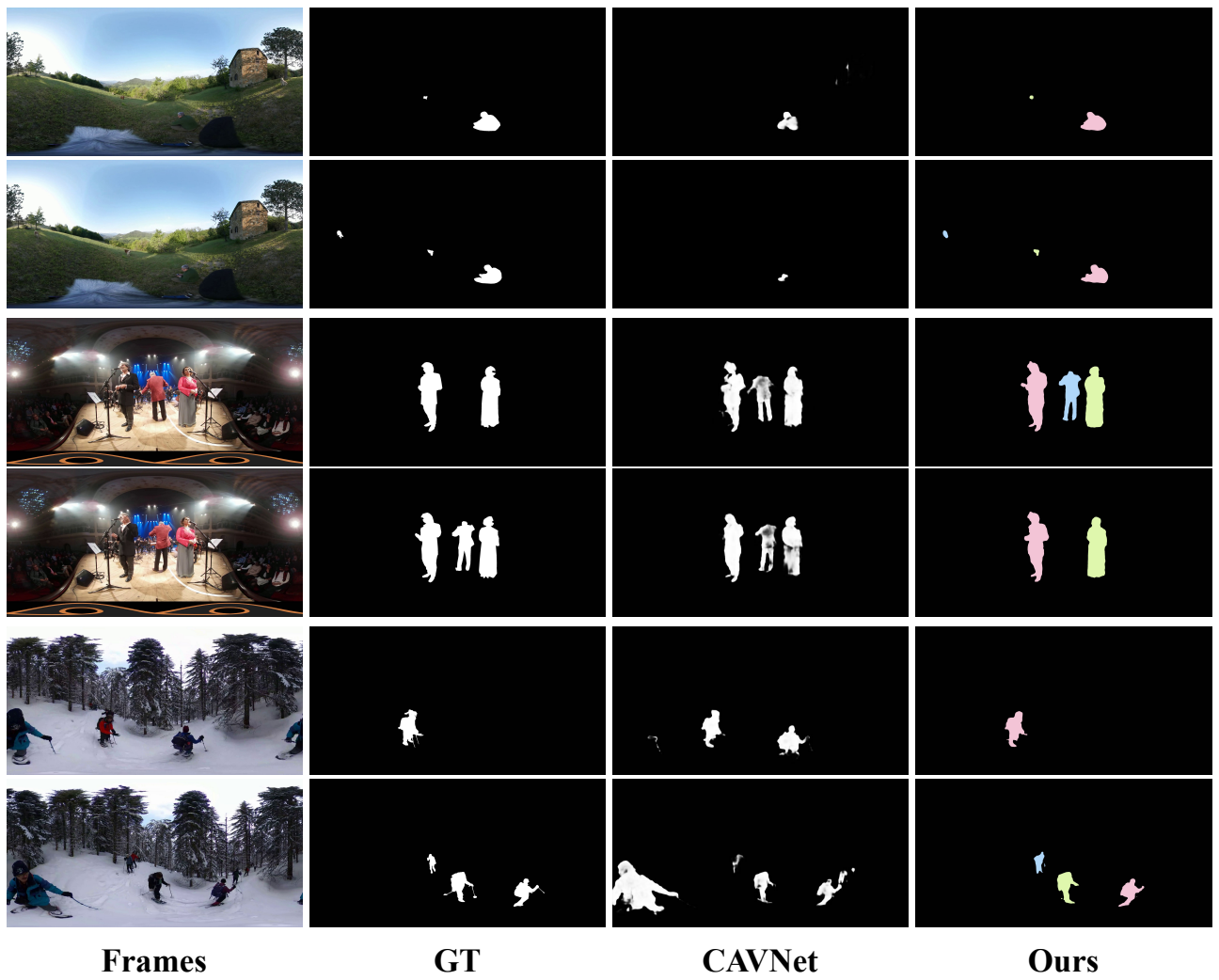

**Figure 6: Predicted saliency results for the *PAVS10K* dataset.**

objects from the multi-instance scenario (e.g., singing persons in the $3^{rd}$ and $4^{th}$ rows); 2) predict the accurate segmentation mask closer to ground truth in severe distortions (e.g., skiing persons in the $6^{th}$ row); 3) detect very small objects (e.g., running dogs in the $2^{nd}$ row). We attribute this to the strong distortion resistance of our DPD and the effective audio-visual fusion of AV-SAM and AV-IAM. Additionally, it is highlighted our method is the first instance-level saliency detection network, and the novelty lies in its ability to differentiate different instances.

## 4.4 Panoramic Audio-Visual Saliency Ranking

**Table 2: Quantitative comparison with different SOR methods on the widely used SOR, #Image used, MAE ($\mathcal{M}$), and adaptive MAE ($\mathcal{AM}$) metrics.**

| Model | SOR ↑ | #Image used ↑ | $\mathcal{M}$ ↓ | $\mathcal{AM}$ ↓ |
|---|---|---|---|---|
| zeros | - | - | **0.020** | 0.460 |
| ASSR [31] | 82.3 | 2477 | 0.034 | 0.316 |
| PSR [33] | **88.5** | **2629** | 0.023 | **0.280** |
| Ours† | **89.5** | **3029** | **0.022** | **0.248** |

To verify the effectiveness of our model on PAV-SOR, we compare our model with ASSR [31] and PSR [33]. Table 2 shows that our model outperforms other recent approaches for all measurements. In panoramic images with 360° omnidirectional view, MAE is dominated by the background region, owing to the small proportion of the foreground region. From the $1^{st}$ row in Table 2, zeros (w/o. masks) achieves the lowest 0.02 MAE. To provide a more comprehensive and unbiased assessment, we introduce a new evaluation metric, namely adaptive MAE ($\mathcal{AM}$), equalizing the significance of foreground and background:

$$\mathcal{AM} = \frac{\sum_{(i,j)\in\mathcal{F}} |G(i,j) - S(i,j)|}{2\mathcal{F}} + \frac{\sum_{(i,j)\in\mathcal{B}} |G(i,j) - S(i,j)|}{2\mathcal{B}} \quad (10)$$

where $G$ and $S$ denote ground truth and the prediction, and $\mathcal{F}$ and $\mathcal{B}$ indicate the foreground and background region of ground truth, respectively. We present our qualitative result in Fig. 7. Compared to PSR [33], our model produces almost-the-same segmentation masks with regard to the ground truth while correctly predicting ranking orders.

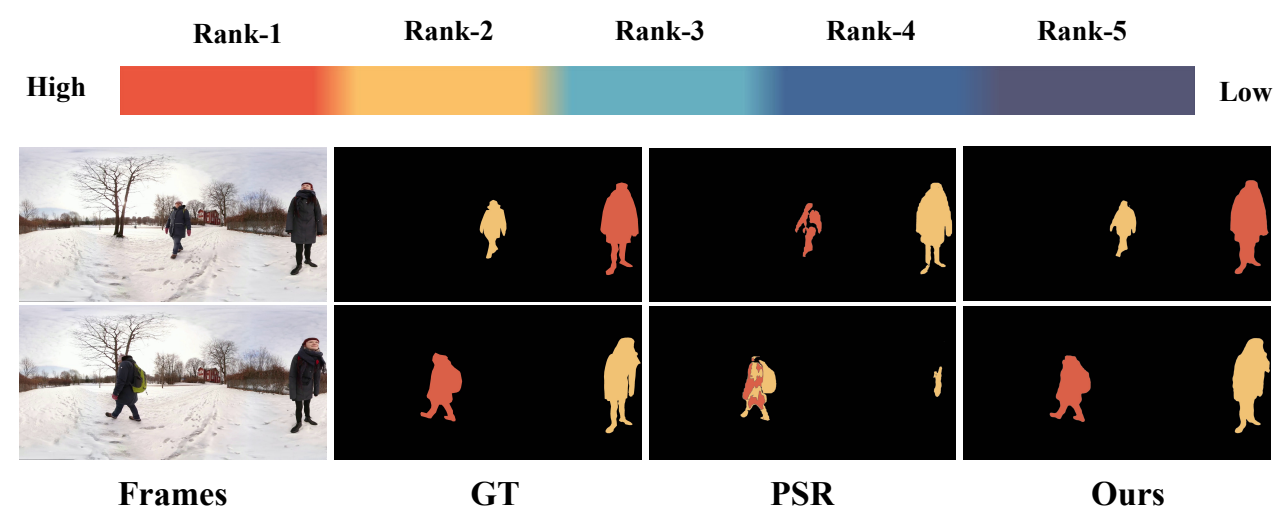

**Figure 7: Qualitative results of rank order of salient objects.**

## 4.5 Ablation Studies

*4.5.1 Impact of Distortion-Aware Pixel Decoder.* Table 3 provides a comprehensive validation of the efficacy of our proposed distortion-aware pixel decoder (DPD). In the $2^{nd}$ row, we introduce DPD, which samples neighbor points only on the equirectangular projection and then aggregates the corresponding neighbor features. The comparison between the first two rows indicates that the integration of local information significantly bolsters the visual feature representation. Nonetheless, the equirectangular map suffers from inherent distortions. The information from rectangular and regular neighbor priors may occasionally deviate from the true spatial arrangement. To address this, we propose cubemap neighbor point sampling, as depicted in Fig. 3. Back-projecting the sampled neighbor points onto the equirectangular map allows us to derive a set of distortion-aware neighbor points. This strategy can effectively capture complex shapes and alleviate the adverse effects of distortions inherent in panoramic images. The results presented in the $3^{rd}$ row underscore the significant performance boost. While the cubemap projection circumvents panoramic distortions, it will induce discontinuity at points near the cube's boundary. Remarkably, by concatenating and jointly utilizing the results derived from both projections, we manage to achieve an optimal outcome, as evidenced in $4^{th}$ row of Table 3.

**Table 3: Impact of DPD and different sampling methods.**

| DPD | Equi. | Cube | $F_\beta$ ↑ | $S_\alpha$ ↑ | $E_\epsilon$ ↑ | $\mathcal{M}$ ↓ |
|---|---|---|---|---|---|---|
| ✘ | ✘ | ✘ | 39.1 | 67.3 | 72.0 | .039 |
| ✔ | ✔ | ✘ | 41.7 | 68.8 | 72.7 | .038 |
| ✔ | ✘ | ✔ | 42.5 | 69.1 | 73.1 | .034 |
| ✔ | ✔ | ✔ | **43.2** | **69.9** | **74.0** | **.033** |

Take a $15 \times 30$ feature map as an example, Fig. 8 visualizes all the central points and their neighbor points that are projected from cube map to equirectangular map. Considering the bottom-left green neighbor point in Fig. 3, its corresponding position is

depicted as a point in the bottom-left map of Fig. 8. To generalize, this bottom-left map in Fig. 8 illustrates the offset position of all bottom-left neighbor points in the original $15 \times 30$ feature map. Similar information is conveyed in the projection maps of the other 7 directions.

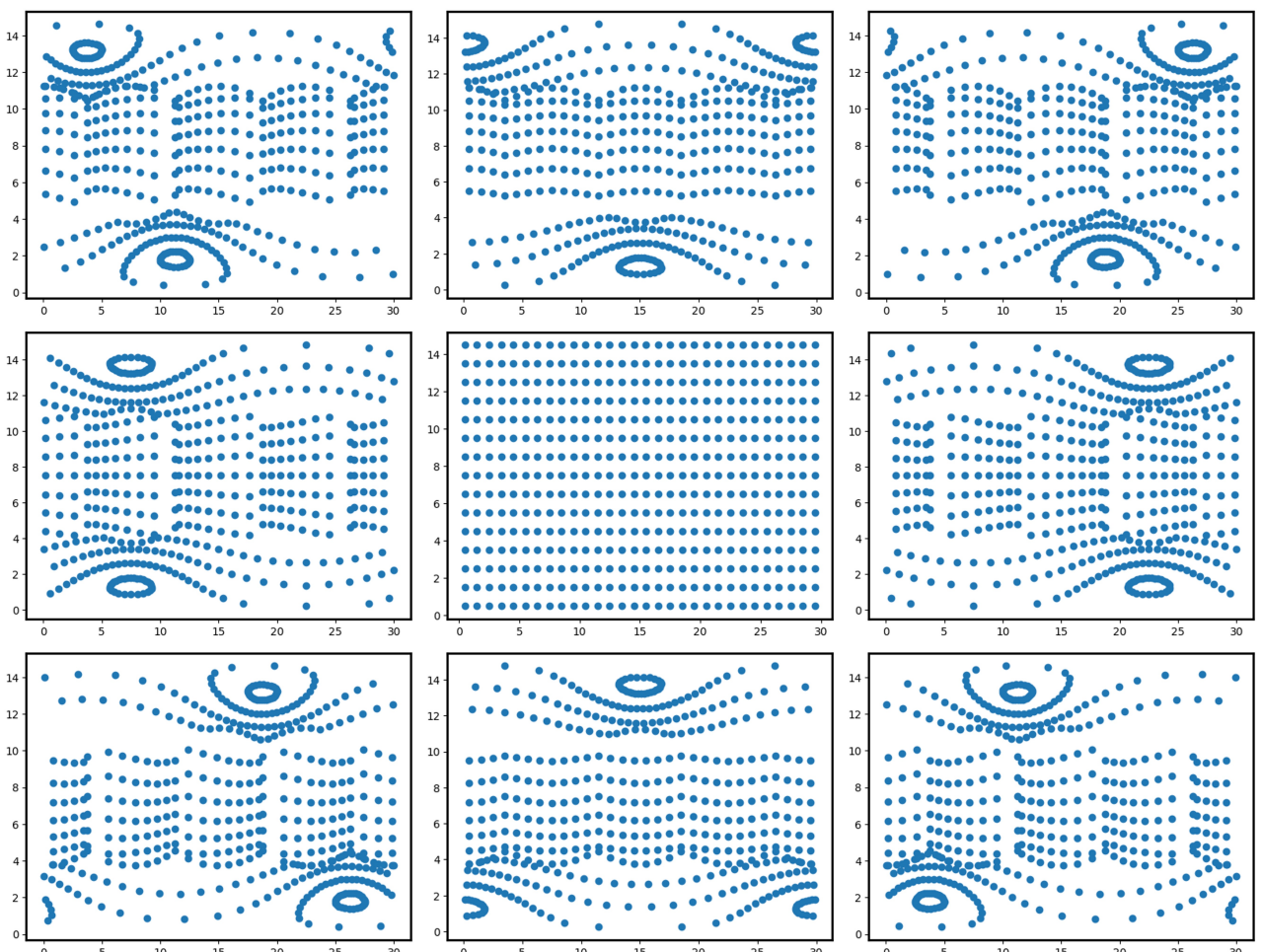

**Figure 8: Visualization of the cube-to-equirectangular projection with respect to the central points and their 8 neighbor points on a $15 \times 30$ feature map.**

*4.5.2 Impact of Sequential Audio-Visual Fusion.* Table 4 validates the efficacy of our proposed sequential audio-visual fusion method, including an audio-visual spatial activation module (AV-SAM) and an audio-visual instance alignment module (AV-IAM). With the integration of AV-SAM, audio signals are seamlessly infused into the visual encoding space via element-wise summation and local convolution operation. It aggregates the audio and image features in a pixel-wise manner, furnishing the pixel decoder with robust localization cues of the sound-emitting object. Row 1 and 2 reveal a noticeable improvement in all metrics, underscoring the crucial role of the acoustic modality in salient object detection and the necessity of our AV-SAM. However, as the sound information of all objects is encompassed within a single audio feature, the model encounters challenges in distinguishing instance-level salient objects. Hence, further unmixing the audio signals is deemed necessary. Row 3 illustrates the performance boost conferred by the AV-IAM, with $F_\beta$ increasing by +1.8%, $S_\alpha$ by +1.0%, and $E_\epsilon$ by +1.3%. This improvement can be attributed to the model's capacity to perform an implicit separation of multiple sound sources, thereby enabling our spatio-temporal object decoder's object queries to possess distinct sound source semantic information. This results in a more efficient and precise frame-by-frame instance-level salient sound source localization. Moreover, we further discuss the cross-modal fusion at various stages. It can be seen that applying the cross-modal fusion at three stages will increase all metrics, showing our model can fuse and balance audio-visual features from multiple stages.

*4.5.3 Impact of Audio-Visual Instance Alignment Module.* Table 5 summarizes the experimental results on the arranging methods of audio-visual and visual-audio transformers. From the results, we can find that performing cross-modal fusion sequentially infers a finer attention map than doing in parallel. Audio prototypes have undergone regularization through contrastive learning before inputting into the visual-audio transformer. Each audio prototype is assigned to a visual object and exhibits heterogeneity with other audio prototypes. Therefore, it helps the visual-audio transformer to align pixels with audio and perform instance-aware fusion.

**Table 4: Impact of sequential audio-visual fusion method.**

| AV-SAM | AV-IAM | level | $F_\beta \uparrow$ | $S_\alpha \uparrow$ | $E_\epsilon \uparrow$ | $\mathcal{M} \downarrow$ |
|---|---|---|---|---|---|---|
| ✘ | ✘ | {3, 4, 5} | 40.8 | 68.3 | 72.2 | .044 |
| ✔ | ✘ | {3, 4, 5} | 41.4 | 68.9 | 72.7 | .034 |
| ✔ | ✔ | {5} | 42.2 | 69.0 | 72.8 | .038 |
| ✔ | ✔ | {3, 4, 5} | **43.2** | **69.9** | **74.0** | **.033** |

**Table 5: Impact of different combining strategies of audio-visual and visual-audio transformer in the AV-IAM.**

| connection | $F_\beta \uparrow$ | $S_\alpha \uparrow$ | $E_\epsilon \uparrow$ | $\mathcal{M} \downarrow$ |
|---|---|---|---|---|
| parallel | 41.9 | 68.6 | 73.5 | .038 |
| sequential | **43.2** | **69.9** | **74.0** | **.033** |

## 4.6 Attention Visualization

To verify that dense audio has been disentangled, we upsample and visualize the attention matrices between audio prototypes and image features from the audio-visual transformer. As illustrated in Fig. 9, when introducing contrastive learning, two audio prototypes can activate two different object instances, that is, the audio signal has been well unmixed and assigned to different sounding objects.

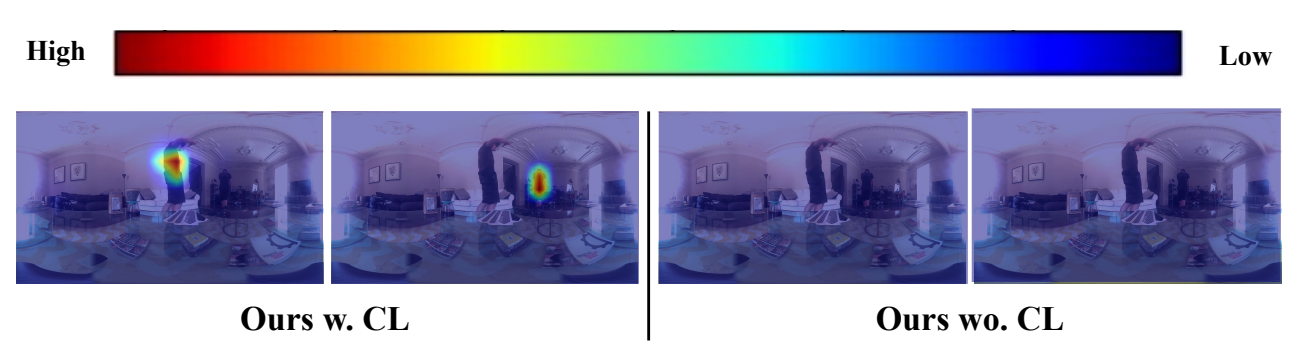

**Figure 9: Attention maps between audio prototypes and image features from the audio-visual transformer in the audio-visual instance alignment module at the fifth stage.**

# 5 CONCLUSION

In this paper, we introduce two challenging tasks, i.e., instance-level PAV-SOD and PAV-SOR, and then propose a unified framework to solve the above tasks. Three key components of our model are designed: a distortion-aware pixel decoder mitigates panoramic distortions; a sequential cross-modal fusion method integrates audio-visual information in an instance-aware manner; a spatio-temporal object decoder generates segmentation mask and saliency rank for each salient object instance. Without bells and whistles, our model achieves satisfactory performances on *PAVS10K* benchmark. We hope the framework could serve as a preferred baseline for panoramic audio-visual saliency detection.

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
