# OpenReview forum: "Instance-Level Panoramic Audio-Visual Saliency Detection and Ranking"
_acmmm.org/ACMMM/2024/Conference — MM2024 Poster_

### Official Review · Reviewer_CfBS · 2024-05-21

**Rating:** 3
**Confidence:** 4

**Summary:**

This paper introduces Instance-level panoramic audio-visual salient object ranking (PAV-SOR), which is a new challenging task. To solve the difficulties in this task, three key modules are designed: a distortion-aware pixel decoder mitigates panoramic distortions, a sequential audio-visual fusion module to integrate audio-visual information; a spatio-temporal object decoder to detect instances and predict scores. Numerous experiments have confirmed the effectiveness of the proposed method.

**Strengths:**

1. The new task proposed in this paper has practical value and provides ranking annotations for PAVS10K dataset, promoting the development of this topic.
2. The distortion-aware pixel decoder is lightweight and efficient, and can serve as the standard preprocessing for panoramic video analysis.

**Limitations:**

1. The author needs to explain how the model annotates the ranking order between instances and how to train the model's ranking ability.
2. I am confused about the correspondence between the two queries f_n^Q and f_n^AV in Eq. (7). It seems that these two terms only correspond based on the index rather than positional embedding. Will this make it difficult for the training of the network?
3. Lines 577 and 563 seem contradictory. Does the symbol f_n^Q represent audio prototypes or query features?

**Suitability:**

2

---

### Official Review · Reviewer_wKDf · 2024-05-24

**Rating:** 4
**Confidence:** 2

**Summary:**

This paper proposes the first instance-level panoramic audio-visual saliency ranking task. A framework with distortion-aware pixel decoding, sequential audio-visual fusion and the spatio-temporal object decoding is proposed and is validated by comprehensive experiments.

**Strengths:**

1. This work introduces the concept of panoramic audio-visual saliency ranking for the first time.
2. A novel method is proposed for panoramic audio-visual saliency detection and ranking in video.
3. The proposed method demonstrates good performance.

**Limitations:**

1. In experiments for panoramic audio-visual saliency ranking, image-based methods are compared, but it is not specified how these methods are applied to videos.
2. In Table 1, the performance of the proposed method in the last column shows no advantage. Further analysis is needed to address this.
3. Although this paper is the first to address instance-level performance, no experiments are presented to demonstrate the method's effectiveness at this level.

**Suitability:**

3

---

### Official Review · Reviewer_LDpt · 2024-05-25

**Rating:** 3
**Confidence:** 3

**Summary:**

This paper primarily focuses on the field of panoramic audio-visual saliency detection and proposes an instance-level panoramic audio-visual saliency detection and ranking framework to detect and rank salient object instances in 360° panoramic videos with audio. The framework designs a distortion-aware pixel decoder to mitigate the inherent geometric distortions in 360° panoramic videos, a sequential cross-modal fusion method to capture instance-aware audio-visual dependencies, and a spatio-temporal object decoder to generate segmentation masks and saliency ranks for each salient object instance.

**Strengths:**

This study proposes an instance-level panoramic audio-visual saliency detection and ranking framework, capable of handling multi-instance saliency detection and ranking tasks simultaneously. The framework effectively mitigates geometric distortions in 360° panoramic videos, achieves efficient audio-visual information fusion, and can separate individual instances and predict their saliency scores. The paper is well-structured, but further validation is needed regarding its practical application prospects.

**Limitations:**

1.Although extensive experiments have been conducted, the comparative experiments are primarily focused on the PAVS10K dataset, lacking comparisons with other types of datasets, which limits the generalizability and applicability of the results.

2.Although the method for generating labels based on attention shift data is introduced, there is a lack of specific construction steps description.

3.The results in Figure 6 are perplexing, with a noticeable misalignment between the predicted results of the third and fourth rows and the labels. For instance, the labels of the third row correspond to the predicted values of the fourth row, and the predictions of the fourth row align with the labels of the third row. The authors did not provide an explanation for this.

4.The explanation of Figure 8 is confusing. It is unclear whether the nine subplots in Figure 8 correspond one-to-one with the green dots in Figure 3. Additionally, there is a need to explain the significance of the points in the figure.

**Suitability:**

2

---

### Official Review · Reviewer_PtoS · 2024-05-25

**Rating:** 4
**Confidence:** 4

**Summary:**

The paper proposes an instance-level framework for segmenting and ranking multiple salient objects in panoramic videos. It introduces three key components: a distortion-aware pixel decoder to handle panoramic distortions, a sequential audio-visual fusion module to integrate audio-visual information, and a spatio-temporal object decoder to separate individual instances and predict their saliency scores. The framework achieves state-of-the-art performance on the PAVS10K benchmark for both saliency detection and ranking tasks .

**Strengths:**

1. The framework introduced in this paper is the first of its kind to address instance-level saliency detection and ranking in panoramic videos.
2. The proposed model achieves satisfactory performances on PAVS10K benchmark.

**Limitations:**

1. As shown in Fig. 1, I'm quite doubtful why the predicted masks of T1 and T2 are different.
2. As in Table 1, why are the performances of the four metrics not consistent? For example, the MAE of COSNet ranks first among all models, but the F-measure is quite limited.
3. In the experiments of saliency ranking, SA-SOR, FLOPS, and FPS are not compared.
4. The necessity of the new metric adaptive MAE is not explained; why not directly use pixel accuracy?
5. The source code is not provided, making some implementation details hard to understand.

If the author can adequately answer these questions, I will raise the score.

**Suitability:**

3

---

### Meta-Review · Area_Chair_MQ9B · 2024-07-03

**Recommendation:** Accept (Poster)
**Confidence:** 4

**Metareview:**

An instance-level audio-visual framework has been proposed for multiple salient object segmentation and ranking simultaneously in panoramic videos. This framework comprises three components: a distortion-aware pixel decoder to mitigate the inherent geometric distortions in 360° panoramic videos, a sequential cross-modal fusion method to capture instance-aware audio-visual dependencies, and a spatio-temporal object decoder to generate segmentation masks and saliency ranks for each salient object instance. The proposed method appears novel and achieves state-of-the-art performance on the PAVS10K benchmark for both saliency detection and ranking tasks. The distortion-aware pixel decoder is lightweight and efficient, and it can serve as standard preprocessing for panoramic video analysis.

The limitations and problems raised by the two reviewers with BR are more related to technical details, equations, etc. Most of the reviewers' concerns have already been addressed in the rebuttal. As stated by the authors in their rebuttal, PAVS10K seems to be the only available dataset for this task. The authors should also incorporate these changes into the paper upon acceptance.